# A Drifting-Games Analysis for Online Learning and Applications to Boosting

**Haipeng Luo**
Department of Computer Science
Princeton University
Princeton, NJ 08540
haipengl@cs.princeton.edu

**Robert E. Schapire**[*]
Department of Computer Science
Princeton University
Princeton, NJ 08540
schapire@cs.princeton.edu

## Abstract

We provide a general mechanism to design online learning algorithms based on a minimax analysis within a drifting-games framework. Different online learning settings (Hedge, multi-armed bandit problems and online convex optimization) are studied by converting into various kinds of drifting games. The original minimax analysis for drifting games is then used and generalized by applying a series of relaxations, starting from choosing a convex surrogate of the 0-1 loss function. With different choices of surrogates, we not only recover existing algorithms, but also propose new algorithms that are totally parameter-free and enjoy other useful properties. Moreover, our drifting-games framework naturally allows us to study high probability bounds without resorting to any concentration results, and also a generalized notion of regret that measures how good the algorithm is compared to all but the top small fraction of candidates. Finally, we translate our new Hedge algorithm into a new adaptive boosting algorithm that is computationally faster as shown in experiments, since it ignores a large number of examples on each round.

## 1   Introduction

In this paper, we study online learning problems within a drifting-games framework, with the aim of developing a general methodology for designing learning algorithms based on a minimax analysis.

To solve an online learning problem, it is natural to consider game-theoretically optimal algorithms which find the best solution even in worst-case scenarios. This is possible for some special cases ([7, 1, 3, 21]) but difficult in general. On the other hand, many other efficient algorithms with optimal regret rate (but not exactly minimax optimal) have been proposed for different learning settings (such as the exponential weights algorithm [14, 15], and follow the perturbed leader [18]). However, it is not always clear how to come up with these algorithms. Recent work by Rakhlin et al. [26] built a bridge between these two classes of methods by showing that many existing algorithms can indeed be derived from a minimax analysis followed by a series of relaxations.

In this paper, we provide a parallel way to design learning algorithms by first converting online learning problems into variants of drifting games, and then applying a minimax analysis and relaxations. *Drifting games* [28] (reviewed in Section 2) generalize Freund's "majority-vote game" [13] and subsume some well-studied boosting and online learning settings. A nearly minimax optimal algorithm is proposed in [28]. It turns out the connections between drifting games and online learning go far beyond what has been discussed previously. To show that, we consider variants of drifting games that capture different popular online learning problems. We then generalize the minimax analysis in [28] based on one key idea: *relax a 0-1 loss function by a convex surrogate*. Although

---

[*]R. Schapire is currently at Microsoft Research in New York City.

this idea has been applied widely elsewhere in machine learning, we use it here in a new way to obtain a very general methodology for designing and analyzing online learning algorithms. Using this general idea, we not only recover existing algorithms, but also design new ones with special useful properties. A somewhat surprising result is that our new algorithms are totally *parameter-free*, which is usually not the case for algorithms derived from a minimax analysis. Moreover, a generalized notion of regret ($\epsilon$-regret, defined in Section 3) that measures how good the algorithm is compared to all but the top $\epsilon$ fraction of candidates arises naturally in our drifting-games framework. Below we summarize our results for a range of learning settings.

**Hedge Settings:** (Section 3) The Hedge problem [14] investigates how to cleverly bet across a set of actions. We show an algorithmic equivalence between this problem and a simple drifting game (DGv1). We then show how to relax the original minimax analysis step by step to reach a general recipe for designing Hedge algorithms (Algorithm 3). Three examples of appropriate convex surrogates of the 0-1 loss function are then discussed, leading to the well-known exponential weights algorithm and two other new ones, one of which (NormalHedge.DT in Section 3.3) bears some similarities with the NormalHedge algorithm [10] and enjoys a similar $\epsilon$-regret bound *simultaneously* for all $\epsilon$ and horizons. However, our regret bounds do not depend on the number of actions, and thus can be applied even when there are infinitely many actions. Our analysis is also arguably simpler and more intuitive than the one in [10] and easy to be generalized to more general settings. Moreover, our algorithm is more computationally efficient since it does not require a numerical searching step as in NormalHedge. Finally, we also derive high probability bounds for the randomized Hedge setting as a simple side product of our framework *without* using any concentration results.

**Multi-armed Bandit Problems:** (Section 4) The multi-armed bandit problem [6] is a classic example for learning with incomplete information where the learner can only obtain feedback for the actions taken. To capture this problem, we study a quite different drifting game (DGv2) where randomness and variance constraints are taken into account. Again the minimax analysis is generalized and the EXP3 algorithm [6] is recovered. Our results could be seen as a preliminary step to answer the open question [2] on exact minimax optimal algorithms for the multi-armed bandit problem.

**Online Convex Optimization:** (Section 4) Based the theory of convex optimization, online convex optimization [31] has been the foundation of modern online learning theory. The corresponding drifting game formulation is a continuous space variant (DGv3). Fortunately, it turns out that all results from the Hedge setting are ready to be used here, recovering the continuous EXP algorithm [12, 17, 24] and also generalizing our new algorithms to this general setting. Besides the usual regret bounds, we also generalize the $\epsilon$-regret, which, as far as we know, is the first time it has been explicitly studied. Again, we emphasize that our new algorithms are adaptive in $\epsilon$ and the horizon.

**Boosting:** (Section 4) Realizing that every Hedge algorithm can be converted into a boosting algorithm ([29]), we propose a new boosting algorithm (NH-Boost.DT) by converting NormalHedge.DT. The adaptivity of NormalHedge.DT is then translated into training error and margin distribution bounds that previous analysis in [29] using nonadaptive algorithms does not show. Moreover, our new boosting algorithm ignores a great many examples on each round, which is an appealing property useful to speeding up the weak learning algorithm. This is confirmed by our experiments.

**Related work**: Our analysis makes use of potential functions. Similar concepts have widely appeared in the literature [8, 5], but unlike our work, they are not related to any minimax analysis and might be hard to interpret. The existence of parameter free Hedge algorithms for unknown number of actions was shown in [11], but no concrete algorithms were given there. Boosting algorithms that ignore some examples on each round were studied in [16], where a heuristic was used to ignore examples with small weights and no theoretical guarantee is provided.

## 2   Reviewing Drifting Games

We consider a simplified version of drifting games similar to the one described in [29, chap. 13] (also called chip games). This game proceeds through $T$ rounds, and is played between a player and an adversary who controls $N$ chips on the real line. The positions of these chips at the end of round $t$ are denoted by $\mathbf{s}_t \in \mathbb{R}^N$, with each coordinate $s_{t,i}$ corresponding to the position of chip $i$. Initially, all chips are at position 0 so that $\mathbf{s}_0 = \mathbf{0}$. On every round $t = 1, \ldots, T$: the player first chooses a distribution $\mathbf{p}_t$ over the chips, then the adversary decides the movements of the chips $\mathbf{z}_t$ so that the

new positions are updated as $\mathbf{s}_t = \mathbf{s}_{t-1} + \mathbf{z}_t$. Here, each $z_{t,i}$ has to be picked from a prespecified set $B \subset \mathbb{R}$, and more importantly, satisfy the constraint $\mathbf{p}_t \cdot \mathbf{z}_t \geq \beta \geq 0$ for some fixed constant $\beta$.

At the end of the game, each chip is associated with a nonnegative loss defined by $L(s_{T,i})$ for some nonincreasing function $L$ mapping from the final position of the chip to $\mathbb{R}_+$. The goal of the player is to minimize the chips' average loss $\frac{1}{N} \sum_{i=1}^{N} L(s_{T,i})$ after $T$ rounds. So intuitively, the player aims to "push" the chips to the right by assigning appropriate weights on them so that the adversary has to move them to the right by $\beta$ in a weighted average sense on each round. This game captures many learning problems. For instance, binary classification via boosting can be translated into a drifting game by treating each training example as a chip (see [28] for details).

We regard a player's strategy $\mathcal{D}$ as a function mapping from the history of the adversary's decisions to a distribution that the player is going to play with, that is, $\mathbf{p}_t = \mathcal{D}(\mathbf{z}_{1:t-1})$ where $\mathbf{z}_{1:t-1}$ stands for $\mathbf{z}_1, \ldots, \mathbf{z}_{t-1}$. The player's worst case loss using this algorithm is then denoted by $L_T(\mathcal{D})$. The minimax optimal loss of the game is computed by the following expression: $\min_{\mathcal{D}} L_T(\mathcal{D}) = \min_{\mathbf{p}_1 \in \Delta_N} \max_{\mathbf{z}_1 \in \mathcal{Z}_{\mathbf{p}_1}} \cdots \min_{\mathbf{p}_T \in \Delta_N} \max_{\mathbf{z}_T \in \mathcal{Z}_{\mathbf{p}_T}} \frac{1}{N} \sum_{i=1}^{N} L(\sum_{t=1}^{T} z_{t,i})$, where $\Delta_N$ is the $N$ dimensional simplex and $\mathcal{Z}_{\mathbf{p}} = B^N \cap \{\mathbf{z} : \mathbf{p} \cdot \mathbf{z} \geq \beta\}$ is assumed to be compact. A strategy $\mathcal{D}^*$ that realizes the minimum in $\min_{\mathcal{D}} L_T(\mathcal{D})$ is called a minimax optimal strategy. A nearly optimal strategy and its analysis is originally given in [28], and a derivation by directly tackling the above minimax expression can be found in [29, chap. 13]. Specifically, a sequence of potential functions of a chip's position is defined recursively as follows:

$$\Phi_T(s) = L(s), \quad \Phi_{t-1}(s) = \min_{w \in \mathbb{R}_+} \max_{z \in B} (\Phi_t(s+z) + w(z-\beta)). \tag{1}$$

Let $w_{t,i}$ be the weight that realizes the minimum in the definition of $\Phi_{t-1}(s_{t-1,i})$, that is, $w_{t,i} \in \arg\min_w \max_z (\Phi_t(s_{t-1,i} + z) + w(z - \beta))$. Then the player's strategy is to set $p_{t,i} \propto w_{t,i}$. The key property of this strategy is that it assures that the sum of the potentials over all the chips never increases, connecting the player's final loss with the potential at time 0 as follows:

$$\frac{1}{N} \sum_{i=1}^{N} L(s_{T,i}) \leq \frac{1}{N} \sum_{i=1}^{N} \Phi_T(s_{T,i}) \leq \frac{1}{N} \sum_{i=1}^{N} \Phi_{T-1}(s_{T-1,i}) \leq \cdots \leq \frac{1}{N} \sum_{i=1}^{N} \Phi_0(s_{0,i}) = \Phi_0(0). \tag{2}$$

It has been shown in [28] that this upper bound on the loss is optimal in a very strong sense.

Moreover, in some cases the potential functions have nice closed forms and thus the algorithm can be efficiently implemented. For example, in the boosting setting, $B$ is simply $\{-1, +1\}$, and one can verify $\Phi_t(s) = \frac{1+\beta}{2} \Phi_{t+1}(s+1) + \frac{1-\beta}{2} \Phi_{t+1}(s-1)$ and $w_{t,i} = \frac{1}{2} (\Phi_t(s_{t-1,i} - 1) - \Phi_t(s_{t-1,i} + 1))$. With the loss function $L(s)$ being $\mathbf{1}\{s \leq 0\}$, these can be further simplified and eventually give exactly the boost-by-majority algorithm [13].

## 3 Online Learning as a Drifting Game

The connection between drifting games and some specific settings of online learning has been noticed before ([28, 23]). We aim to find deeper connections or even an equivalence between variants of drifting games and more general settings of online learning, and provide insights on designing learning algorithms through a minimax analysis. We start with a simple yet classic Hedge setting.

### 3.1 Algorithmic Equivalence

In the Hedge setting [14], a player tries to earn as much as possible (or lose as little as possible) by cleverly spreading a fixed amount of money to bet on a set of actions on each day. Formally, the game proceeds for $T$ rounds, and on each round $t = 1, \ldots, T$: the player chooses a distribution $\mathbf{p}_t$ over $N$ actions, then the adversary decides the actions' losses $\boldsymbol{\ell}_t$ (i.e. action $i$ incurs loss $\ell_{t,i} \in [0, 1]$) which are revealed to the player. The player suffers a weighted average loss $\mathbf{p}_t \cdot \boldsymbol{\ell}_t$ at the end of this round. The goal of the player is to minimize his "regret", which is usually defined as the difference between his total loss and the loss of the best action. Here, we consider an even more general notion of regret studied in [20, 19, 10, 11], which we call $\epsilon$-*regret*. Suppose the actions are ordered according to their total losses after $T$ rounds (i.e. $\sum_{t=1}^{T} \ell_{t,i}$) from smallest to largest, and let $i_\epsilon$ be the index

| **Input**: A Hedge Algorithm $\mathcal{H}$ | **Input**: A DGv1 Algorithm $\mathcal{D}_R$ |
|---|---|
| **for** $t = 1$ **to** $T$ **do** | **for** $t = 1$ **to** $T$ **do** |
|    Query $\mathcal{H}$: $\mathbf{p}_t = \mathcal{H}(\boldsymbol{\ell}_{1:t-1})$. |    Query $\mathcal{D}_R$: $\mathbf{p}_t = \mathcal{D}_R(\mathbf{z}_{1:t-1})$. |
|    Set: $\mathcal{D}_R(\mathbf{z}_{1:t-1}) = \mathbf{p}_t$. |    Set: $\mathcal{H}(\boldsymbol{\ell}_{1:t-1}) = \mathbf{p}_t$. |
|    Receive movements $\mathbf{z}_t$ from the adversary. |    Receive losses $\boldsymbol{\ell}_t$ from the adversary. |
|    Set: $\ell_{t,i} = z_{t,i} - \min_j z_{t,j}, \ \forall i$. |    Set: $z_{t,i} = \ell_{t,i} - \mathbf{p}_t \cdot \boldsymbol{\ell}_t, \ \forall i$. |

**Algorithm 1:** Conversion of a Hedge Algorithm $\mathcal{H}$ to a DGv1 Algorithm $\mathcal{D}_R$      **Algorithm 2:** Conversion of a DGv1 Algorithm $\mathcal{D}_R$ to a Hedge Algorithm $\mathcal{H}$

of the action that is the $\lceil N\epsilon \rceil$-th element in the sorted list ($0 < \epsilon \le 1$). Now, $\epsilon$-regret is defined as $\mathbf{R}_T^\epsilon(\mathbf{p}_{1:T}, \boldsymbol{\ell}_{1:T}) = \sum_{t=1}^T \mathbf{p}_t \cdot \boldsymbol{\ell}_t - \sum_{t=1}^T \ell_{t,i_\epsilon}$. In other words, $\epsilon$-regret measures the difference between the player's loss and the loss of the $\lceil N\epsilon \rceil$-th best action (recovering the usual regret with $\epsilon \le 1/N$), and sublinear $\epsilon$-regret implies that the player's loss is almost as good as all but the top $\epsilon$ fraction of actions. Similarly, $\mathbf{R}_T^\epsilon(\mathcal{H})$ denotes the worst case $\epsilon$-regret for a specific algorithm $\mathcal{H}$. For convenience, when $\epsilon \le 0$ or $\epsilon > 1$, we define $\epsilon$-regret to be $\infty$ or $-\infty$ respectively.

Next we discuss how Hedge is highly related to drifting games. Consider a variant of drifting games where $B = [-1,1], \beta = 0$ and $L(s) = \mathbf{1}\{s \le -R\}$ for some constant $R$. Additionally, we impose an extra restriction on the adversary: $|z_{t,i} - z_{t,j}| \le 1$ for all $i$ and $j$. In other words, the difference between any two chips' movements is at most 1. We denote this specific variant of drifting games by DGv1 (summarized in Appendix A) and a corresponding algorithm by $\mathcal{D}_R$ to emphasize the dependence on $R$. The reductions in Algorithm 1 and 2 and Theorem 1 show that DGv1 and the Hedge problem are algorithmically equivalent (note that both conversions are valid). The proof is straightforward and deferred to Appendix B. By Theorem 1, it is clear that the minimax optimal algorithm for one setting is also minimax optimal for the other under these conversions.

**Theorem 1.** *DGv1 and the Hedge problem are algorithmically equivalent in the following sense:*
*(1) Algorithm 1 produces a DGv1 algorithm $\mathcal{D}_R$ satisfying $L_T(\mathcal{D}_R) \le i/N$ where $i \in \{0, \ldots, N\}$ is such that $\mathbf{R}_T^{(i+1)/N}(\mathcal{H}) < R \le \mathbf{R}_T^{i/N}(\mathcal{H})$.*

*(2) Algorithm 2 produces a Hedge algorithm $\mathcal{H}$ with $\mathbf{R}_T^\epsilon(\mathcal{H}) < R$ for any $R$ such that $L_T(\mathcal{D}_R) < \epsilon$.*

### 3.2 Relaxations

From now on we only focus on the direction of converting a drifting game algorithm into a Hedge algorithm. In order to derive a minimax Hedge algorithm, Theorem 1 tells us it suffices to derive minimax DGv1 algorithms. Exact minimax analysis is usually difficult, and appropriate relaxations seem to be necessary. To make use of the existing analysis for standard drifting games, the first obvious relaxation is to drop the additional restriction in DGv1, that is, $|z_{t,i} - z_{t,j}| \le 1$ for all $i$ and $j$. Doing this will lead to the exact setting discussed in [23] where a near optimal strategy is proposed using the recipe in Eq. (1). It turns out that this relaxation is reasonable and does not give too much more power to the adversary. To see this, first recall that results from [23], written in our notation, state that $\min_{\mathcal{D}_R} L_T(\mathcal{D}_R) \le \frac{1}{2^T} \sum_{j=0}^{\frac{T-R}{2}} \binom{T+1}{j}$, which, by Hoeffding's inequality, is upper bounded by $2\exp\left(-\frac{(R+1)^2}{2(T+1)}\right)$. Second, statement (2) in Theorem 1 clearly remains valid if the input of Algorithm 2 is a drifting game algorithm for this relaxed version of DGv1. Therefore, by setting $\epsilon > 2\exp\left(-\frac{(R+1)^2}{2(T+1)}\right)$ and solving for $R$, we have $\mathbf{R}_T^\epsilon(\mathcal{H}) \le O\left(\sqrt{T\ln(\frac{1}{\epsilon})}\right)$, which is the known optimal regret rate for the Hedge problem, showing that we lose little due to this relaxation.

However, the algorithm proposed in [23] is not computationally efficient since the potential functions $\Phi_t(s)$ do not have closed forms. To get around this, we would want the minimax expression in Eq. (1) to be easily solved, just like the case when $B = \{-1, 1\}$. It turns out that convexity would allow us to treat $B = [-1, 1]$ almost as $B = \{-1, 1\}$. Specifically, if each $\Phi_t(s)$ is a convex function of $s$, then due to the fact that the maximum of a convex function is always realized at the boundary of a compact region, we have

$$\min_{w \in \mathbb{R}_+} \max_{z \in [-1,1]} \left(\Phi_t(s+z) + wz\right) = \min_{w \in \mathbb{R}_+} \max_{z \in \{-1,1\}} \left(\Phi_t(s+z) + wz\right) = \frac{\Phi_t(s-1) + \Phi_t(s+1)}{2},$$

$$(3)$$

---

**Input**: A convex, nonincreasing, nonnegative function $\Phi_T(s)$.
**for** $t = T$ **down to** $1$ **do**
    Find a convex function $\Phi_{t-1}(s)$ s.t. $\forall s,\ \Phi_t(s-1) + \Phi_t(s+1) \leq 2\Phi_{t-1}(s)$.
Set: $\mathbf{s}_0 = \mathbf{0}$.
**for** $t = 1$ **to** $T$ **do**
    Set: $\mathcal{H}(\boldsymbol{\ell}_{1:t-1}) = \mathbf{p}_t$ s.t. $p_{t,i} \propto \Phi_t(s_{t-1,i} - 1) - \Phi_t(s_{t-1,i} + 1)$.
    Receive losses $\boldsymbol{\ell}_t$ and set $s_{t,i} = s_{t-1,i} + \ell_{t,i} - \mathbf{p}_t \cdot \boldsymbol{\ell}_t,\ \forall i$.

---

**Algorithm 3:** A General Hedge Algorithm $\mathcal{H}$

with $w = (\Phi_t(s-1) - \Phi_t(s+1))/2$ realizing the minimum. Since the 0-1 loss function $L(s)$ is not convex, this motivates us to find a convex surrogate of $L(s)$. Fortunately, relaxing the equality constraints in Eq. (1) does not affect the key property of Eq. (2) as we will show in the proof of Theorem 2. "Compiling out" the input of Algorithm 2, we thus have our general recipe (Algorithm 3) for designing Hedge algorithms with the following regret guarantee.

**Theorem 2.** *For Algorithm 3, if $R$ and $\epsilon$ are such that $\Phi_0(0) < \epsilon$ and $\Phi_T(s) \geq \mathbf{1}\{s \leq -R\}$ for all $s \in \mathbb{R}$, then $\mathbf{R}_T^\epsilon(\mathcal{H}) < R$.*

*Proof.* It suffices to show that Eq. (2) holds so that the theorem follows by a direct application of statement (2) of Theorem 1. Let $w_{t,i} = (\Phi_t(s_{t-1,i} - 1) - \Phi_t(s_{t-1,i} + 1))/2$. Then $\sum_i \Phi_t(s_{t,i}) \leq \sum_i (\Phi_t(s_{t-1,i} + z_{t,i}) + w_{t,i} z_{t,i})$ since $p_{t,i} \propto w_{t,i}$ and $\mathbf{p}_t \cdot \mathbf{z}_t \geq 0$. On the other hand, by Eq. (3), we have $\Phi_t(s_{t-1,i} + z_{t,i}) + w_{t,i} z_{t,i} \leq \min_{w \in \mathbb{R}_+} \max_{z \in [-1,1]} (\Phi_t(s_{t-1,i} + z) + wz) = \frac{1}{2}(\Phi_t(s_{t-1,i} - 1) + \Phi_t(s_{t-1,i} + 1))$, which is at most $\Phi_{t-1}(s_{t-1,i})$ by Algorithm 3. This shows $\sum_i \Phi_t(s_{t,i}) \leq \sum_i \Phi_{t-1}(s_{t-1,i})$ and Eq. (2) follows. $\square$

Theorem 2 tells us that if solving $\Phi_0(0) < \epsilon$ for $R$ gives $R > \underline{R}$ for some value $\underline{R}$, then the regret of Algorithm 3 is less than any value that is greater than $\underline{R}$, meaning the regret is at most $\underline{R}$.

### 3.3 Designing Potentials and Algorithms

Now we are ready to recover existing algorithms and develop new ones by choosing an appropriate potential $\Phi_T(s)$ as Algorithm 3 suggests. We will discuss three different algorithms below, and summarize these examples in Table 1 (see Appendix C).

**Exponential Weights (EXP) Algorithm.** Exponential loss is an obvious choice for $\Phi_T(s)$ as it has been widely used as the convex surrogate of the 0-1 loss function in the literature. It turns out that this will lead to the well-known exponential weights algorithm [14, 15]. Specifically, we pick $\Phi_T(s)$ to be $\exp(-\eta(s+R))$ which exactly upper bounds $\mathbf{1}\{s \leq -R\}$. To compute $\Phi_t(s)$ for $t \leq T$, we simply let $\Phi_t(s-1) + \Phi_t(s+1) \leq 2\Phi_{t-1}(s)$ hold with equality. Indeed, direct computations show that all $\Phi_t(s)$ share a similar form: $\Phi_t(s) = \left(\frac{e^\eta + e^{-\eta}}{2}\right)^{T-t} \cdot \exp(-\eta(s+R))$. Therefore, according to Algorithm 3, the player's strategy is to set

$$p_{t,i} \propto \Phi_t(s_{t-1,i} - 1) - \Phi_t(s_{t-1,i} + 1) \propto \exp(-\eta s_{t-1,i}),$$

which is exactly the same as EXP (note that $R$ becomes irrelevant after normalization). To derive regret bounds, it suffices to require $\Phi_0(0) < \epsilon$, which is equivalent to $R > \frac{1}{\eta}\left(\ln(\frac{1}{\epsilon}) + T \ln \frac{e^\eta + e^{-\eta}}{2}\right)$. By Theorem 2 and Hoeffding's lemma (see [9, Lemma A.1]), we thus know $\mathbf{R}_T^\epsilon(\mathcal{H}) \leq \frac{1}{\eta}\ln\left(\frac{1}{\epsilon}\right) + \frac{T\eta}{2} = \sqrt{2T \ln\left(\frac{1}{\epsilon}\right)}$ where the last step is by optimally tuning $\eta$ to be $\sqrt{2(\ln\frac{1}{\epsilon})/T}$. Note that this algorithm is *not adaptive* in the sense that it requires knowledge of $T$ and $\epsilon$ to set the parameter $\eta$.

We have thus recovered the well-known EXP algorithm and given a new analysis using the drifting-games framework. More importantly, as in [26], this derivation may shed light on why this algorithm works and where it comes from, namely, a minimax analysis followed by a series of relaxations, starting from a reasonable surrogate of the 0-1 loss function.

**2-norm Algorithm.** We next move on to another simple convex surrogate: $\Phi_T(s) = a[s]_-^2 \geq \mathbf{1}\{s \leq -1/\sqrt{a}\}$, where $a$ is some positive constant and $[s]_- = \min\{0, s\}$ represents a truncating operation. The following lemma shows that $\Phi_t(s)$ can also be simply described.

**Lemma 1.** *If $a > 0$, then $\Phi_t(s) = a\left([s]_-^2 + T - t\right)$ satisfies $\Phi_t(s-1) + \Phi_t(s+1) \leq 2\Phi_{t-1}(s)$.*

Thus, Algorithm 3 can again be applied. The resulting algorithm is extremely concise:

$$p_{t,i} \propto \Phi_t(s_{t-1,i} - 1) - \Phi_t(s_{t-1,i} + 1) \propto [s_{t-1,i} - 1]_-^2 - [s_{t-1,i} + 1]_-^2.$$

We call this the "2-norm" algorithm since it resembles the $p$-norm algorithm in the literature when $p = 2$ (see [9]). The difference is that the $p$-norm algorithm sets the weights proportional to the derivative of potentials, instead of the difference of them as we are doing here. A somewhat surprising property of this algorithm is that it is totally adaptive and parameter-free (since $a$ disappears under normalization), a property that we usually do not expect to obtain from a minimax analysis. Direct application of Theorem 2 ($\Phi_0(0) = aT < \epsilon \Leftrightarrow 1/\sqrt{a} > \sqrt{T/\epsilon}$) shows that its regret achieves the optimal dependence on the horizon $T$.

**Corollary 1.** *Algorithm 3 with potential $\Phi_t(s)$ defined in Lemma 1 produces a Hedge algorithm $\mathcal{H}$ such that $\mathbf{R}_T^\epsilon(\mathcal{H}) \leq \sqrt{T/\epsilon}$ simultaneously for all $T$ and $\epsilon$.*

**NormalHedge.DT.** The regret for the 2-norm algorithm does not have the optimal dependence on $\epsilon$. An obvious follow-up question would be whether it is possible to derive an adaptive algorithm that achieves the optimal rate $O(\sqrt{T\ln(1/\epsilon)})$ simultaneously for all $T$ and $\epsilon$ using our framework. An even deeper question is: instead of choosing convex surrogates in a seemingly arbitrary way, is there a more natural way to find the *right* choice of $\Phi_T(s)$?

To answer these questions, we recall that the reason why the 2-norm algorithm can get rid of the dependence on $\epsilon$ is that $\epsilon$ appears merely in the multiplicative constant $a$ that does not play a role after normalization. This motivates us to let $\Phi_T(s)$ in the form of $\epsilon F(s)$ for some $F(s)$. On the other hand, from Theorem 2, we also want $\epsilon F(s)$ to upper bound the 0-1 loss function $\mathbf{1}\{s \leq -\sqrt{dT\ln(1/\epsilon)}\}$ for some constant $d$. Taken together, this is telling us that the right choice of $F(s)$ should be of the form $\Theta\left(\exp(s^2/T)\right)$[1]. Of course we still need to refine it to satisfy the monotonicity and other properties. We define $\Phi_T(s)$ formally and more generally as:

$$\Phi_T(s) = a\left(\exp\left(\frac{[s]_-^2}{dT}\right) - 1\right) \geq \mathbf{1}\left\{s \leq -\sqrt{dT\ln\left(\frac{1}{a} + 1\right)}\right\},$$

where $a$ and $d$ are some positive constants. This time it is more involved to figure out what other $\Phi_t(s)$ should be. The following lemma addresses this issue (proof deferred to Appendix C).

**Lemma 2.** *If $b_t = 1 - \frac{1}{2}\sum_{\tau=t+1}^T \left(\exp\left(\frac{4}{d\tau}\right) - 1\right), a > 0, d \geq 3$ and $\Phi_t(s) = a\left(\exp\left(\frac{[s]_-^2}{dt}\right) - b_t\right)$ (define $\Phi_0(s) \equiv a(1 - b_0)$), then we have $\Phi_t(s-1) + \Phi_t(s+1) \leq 2\Phi_{t-1}(s)$ for all $s \in \mathbb{R}$ and $t = 2, \ldots, T$. Moreover, Eq. (2) still holds.*

Note that even if $\Phi_1(s-1) + \Phi_1(s+1) \leq 2\Phi_0(s)$ is not valid in general, Lemma 2 states that Eq. (2) still holds. Thus Algorithm 3 can indeed still be applied, leading to our new algorithm:

$$p_{t,i} \propto \Phi_t(s_{t-1,i} - 1) - \Phi_t(s_{t-1,i} + 1) \propto \exp\left(\frac{[s_{t-1,i} - 1]_-^2}{dt}\right) - \exp\left(\frac{[s_{t-1,i} + 1]_-^2}{dt}\right).$$

Here, $d$ seems to be an extra parameter, but in fact, simply setting $d = 3$ is good enough:

**Corollary 2.** *Algorithm 3 with potential $\Phi_t(s)$ defined in Lemma 2 and $d = 3$ produces a Hedge algorithm $\mathcal{H}$ such that the following holds simultaneously for all $T$ and $\epsilon$:*

$$\mathbf{R}_T^\epsilon(\mathcal{H}) \leq \sqrt{3T\ln\left(\frac{1}{2\epsilon}\left(e^{4/3} - 1\right)(\ln T + 1) + 1\right)} = O\left(\sqrt{T\ln(1/\epsilon) + T\ln\ln T}\right).$$

We have thus proposed a parameter-free adaptive algorithm with optimal regret rate (ignoring the $\ln\ln T$ term) using our drifting-games framework. In fact, our algorithm bears a striking similarity to NormalHedge [10], the first algorithm that has this kind of adaptivity. We thus name our algorithm NormalHedge.DT[2]. We include NormalHedge in Table 1 for comparison. One can see that the main differences are: 1) On each round NormalHedge performs a numerical search to find out the right parameter used in the exponents; 2) NormalHedge uses the derivative of potentials as weights.

Compared to NormalHedge, the regret bound for NormalHedge.DT has no explicit dependence on $N$, but has a slightly worse dependence on $T$ (indeed $\ln\ln T$ is almost negligible). We emphasize other advantages of our algorithm over NormalHedge: 1) NormalHedge.DT is more computationally efficient especially when $N$ is very large, since it does not need a numerical search for each round; 2) our analysis is arguably simpler and more intuitive than the one in [10]; 3) as we will discuss in Section 4, NormalHedge.DT can be easily extended to deal with the more general online convex optimization problem where the number of actions is infinitely large, while it is not clear how to do that for NormalHedge by generalizing the analysis in [10]. Indeed, the extra dependence on the number of actions $N$ for the regret of NormalHedge makes this generalization even seem impossible. Finally, we will later see that NormalHedge.DT outperforms NormalHedge in experiments. Despite the differences, it is worth noting that both algorithms assign zero weight to some actions on each round, an appealing property when $N$ is huge. We will discuss more on this in Section 4.

### 3.4 High Probability Bounds

We now consider a common variant of Hedge: on each round, instead of choosing a distribution $\mathbf{p}_t$, the player has to randomly pick a single action $i_t$, while the adversary decides the losses $\boldsymbol{\ell}_t$ at the same time (without seeing $i_t$). For now we only focus on the player's regret to the best action: $\mathbf{R}_T(i_{1:T}, \boldsymbol{\ell}_{1:T}) = \sum_{t=1}^{T} \ell_{t,i_t} - \min_i \sum_{t=1}^{T} \ell_{t,i}$. Notice that the regret is now a random variable, and we are interested in a bound that holds with high probability. Using Azuma's inequality, standard analysis (see for instance [9, Lemma 4.1]) shows that the player can simply draw $i_t$ according to $\mathbf{p}_t = \mathcal{H}(\boldsymbol{\ell}_{1:t-1})$, the output of a standard Hedge algorithm, and suffers regret at most $\mathbf{R}_T(\mathcal{H}) + \sqrt{T\ln(1/\delta)}$ with probability $1 - \delta$. Below we recover similar results as a simple side product of our drifting-games analysis *without* resorting to concentration results, such as Azuma's inequality.

For this, we only need to modify Algorithm 3 by setting $z_{t,i} = \ell_{t,i} - \ell_{t,i_t}$. The restriction $\mathbf{p}_t \cdot \mathbf{z}_t \geq 0$ is then relaxed to hold in expectation. Moreover, it is clear that Eq. (2) also still holds in expectation. On the other hand, by definition and the union bound, one can show that $\sum_i \mathbb{E}[L(s_{T,i})] = \sum_i \Pr\left[s_{T,i} \leq -R\right] \geq \Pr\left[\mathbf{R}_T(i_{1:T}, \boldsymbol{\ell}_{1:T}) \geq R\right]$. So setting $\Phi_0(0) = \delta$ shows that the regret is smaller than $R$ with probability $1 - \delta$. Therefore, for example, if EXP is used, then the regret would be at most $\sqrt{2T\ln(N/\delta)}$ with probability $1 - \delta$, giving basically the same bound as the standard analysis. One draw back is that EXP would need $\delta$ as a parameter. However, this can again be addressed by NormalHedge.DT for the exact same reason that NormalHedge.DT is independent of $\epsilon$. We have thus derived high probability bounds without using any concentration inequalities.

## 4 Generalizations and Applications

**Multi-armed Bandit (MAB) Problem:** The only difference between Hedge (randomized version) and the non-stochastic MAB problem [6] is that on each round, after picking $i_t$, the player only sees the loss for this single action $\ell_{t,i_t}$ instead of the whole vector $\boldsymbol{\ell}_t$. The goal is still to compete with the best action. A common technique used in the bandit setting is to build an unbiased estimator $\hat{\boldsymbol{\ell}}_t$ for the losses, which in this case could be $\hat{\ell}_{t,i} = \mathbf{1}\{i = i_t\} \cdot \ell_{t,i_t}/p_{t,i_t}$. Then algorithms such as EXP can be used by replacing $\boldsymbol{\ell}_t$ with $\hat{\boldsymbol{\ell}}_t$, leading to the EXP3 algorithm [6] with regret $O(\sqrt{TN\ln N})$.

One might expect that Algorithm 3 would also work well by replacing $\boldsymbol{\ell}_t$ with $\hat{\boldsymbol{\ell}}_t$. However, doing so breaks an important property of the movements $z_{t,i}$: boundedness. Indeed, Eq. (3) no longer makes sense if $z$ could be infinitely large, even if in expectation it is still in $[-1, 1]$ (note that $z_{t,i}$ is now a random variable). It turns out that we can address this issue by imposing a variance constraint on $z_{t,i}$. Formally, we consider a variant of drifting games where on each round, the adversary picks a random movement $z_{t,i}$ for each chip such that: $z_{t,i} \geq -1$, $\mathbb{E}_t[z_{t,i}] \leq 1$, $\mathbb{E}_t[z_{t,i}^2] \leq 1/p_{t,i}$ and $\mathbb{E}_t[\mathbf{p}_t \cdot \mathbf{z}_t] \geq 0$. We call this variant DGv2 and summarize it in Appendix A. The standard minimax analysis and the derivation of potential functions need to be modified in a certain way for DGv2, as stated in Theorem 4 (Appendix D). Using the analysis for DGv2, we propose a general recipe for designing MAB algorithms in a similar way as for Hedge and also recover EXP3 (see Algorithm 4 and Theorem 5 in Appendix D). Unfortunately so far we do not know other appropriate potentials due to some technical difficulties. We conjecture, however, that there is a potential function that could recover the poly-INF algorithm [4, 5] or give its variants that achieve the optimal regret $O(\sqrt{TN})$.

**Online Convex Optimization:** We next consider a general online convex optimization setting [31]. Let $S \subset \mathbb{R}^d$ be a compact convex set, and $\mathcal{F}$ be a set of convex functions with range $[0, 1]$ on $S$. On each round $t$, the learner chooses a point $\mathbf{x}_t \in S$, and the adversary chooses a loss function $f_t \in \mathcal{F}$ (knowing $\mathbf{x}_t$). The learner then suffers loss $f_t(\mathbf{x}_t)$. The regret after $T$ rounds is $\mathbf{R}_T(\mathbf{x}_{1:T}, f_{1:T}) = \sum_{t=1}^{T} f_t(\mathbf{x}_t) - \min_{\mathbf{x} \in S} \sum_{t=1}^{T} f_t(\mathbf{x})$. There are two general approaches to OCO: one builds on convex optimization theory [30], and the other generalizes EXP to a continuous space [12, 24]. We will see how the drifting-games framework can recover the latter method and also leads to new ones.

To do so, we introduce a continuous variant of drifting games (DGv3, see Appendix A). There are now infinitely many chips, one for each point in $S$. On round $t$, the player needs to choose a distribution over the chips, that is, a probability density function $p_t(\mathbf{x})$ on $S$. Then the adversary decides the movements for each chip, that is, a function $z_t(\mathbf{x})$ with range $[-1, 1]$ on $S$ (not necessarily convex or continuous), subject to a constraint $\mathbb{E}_{\mathbf{x} \sim p_t}[z_t(\mathbf{x})] \geq 0$. At the end, each point $\mathbf{x}$ is associated with a loss $L(\mathbf{x}) = \mathbf{1}\{\sum_t z_t(\mathbf{x}) \leq -R\}$, and the player aims to minimize the total loss $\int_{\mathbf{x} \in S} L(\mathbf{x}) d\mathbf{x}$.

OCO can be converted into DGv3 by setting $z_t(\mathbf{x}) = f_t(\mathbf{x}) - f_t(\mathbf{x}_t)$ and predicting $\mathbf{x}_t = \mathbb{E}_{\mathbf{x} \sim p_t}[\mathbf{x}] \in S$. The constraint $\mathbb{E}_{\mathbf{x} \sim p_t}[z_t(\mathbf{x})] \geq 0$ holds by the convexity of $f_t$. Moreover, it turns out that the minimax analysis and potentials for DGv1 can readily be used here, and the notion of $\epsilon$-regret, now generalized to the OCO setting, measures the difference of the player's loss and the loss of a best fixed point in a subset of $S$ that excludes the top $\epsilon$ fraction of points. With different potentials, we obtain versions of each of the three algorithms of Section 3 generalized to this setting, with the same $\epsilon$-regret bounds as before. Again, two of these methods are adaptive and parameter-free. To derive bounds for the usual regret, at first glance it seems that we have to set $\epsilon$ to be close to zero, leading to a meaningless bound. Nevertheless, this is addressed by Theorem 6 using similar techniques in [17], giving the usual $O(\sqrt{dT \ln T})$ regret bound. All details can be found in Appendix E.

**Applications to Boosting:** There is a deep and well-known connection between Hedge and boosting [14, 29]. In principle, every Hedge algorithm can be converted into a boosting algorithm; for instance, this is how AdaBoost was derived from EXP. In the same way, NormalHedge.DT can be converted into a new boosting algorithm that we call NH-Boost.DT. See Appendix F for details and further background on boosting. The main idea is to treat each training example as an "action", and to rely on the Hedge algorithm to compute distributions over these examples which are used to train the weak hypotheses. Typically, it is assumed that each of these has "edge" $\gamma$, meaning its accuracy on the training distribution is at least $1/2 + \gamma$. The final hypothesis is a simple majority vote of the weak hypotheses. To understand the prediction accuracy of a boosting algorithm, we often study the training error rate and also the distribution of margins, a well-established measure of confidence (see Appendix F for formal definitions). Thanks to the adaptivity of NormalHedge.DT, we can derive bounds on both the training error and the distribution of margins after any number of rounds:

**Theorem 3.** *After $T$ rounds, the training error of NH-Boost.DT is of order $\tilde{O}(\exp(-\frac{1}{3}T\gamma^2))$, and the fraction of training examples with margin at most $\theta (\leq 2\gamma)$ is of order $\tilde{O}(\exp(-\frac{1}{3}T(\theta - 2\gamma)^2))$.*

Thus, the training error decreases at roughly the same rate as AdaBoost. In addition, this theorem implies that the fraction of examples with margin smaller than $2\gamma$ eventually goes to zero as $T$ gets large, which means NH-Boost.DT converges to the optimal margin $2\gamma$; this is known not to be true for AdaBoost (see [29]). Also, like AdaBoost, NH-Boost.DT is an adaptive boosting algorithm that does not require $\gamma$ or $T$ as a parameter. However, unlike AdaBoost, NH-Boost.DT has the striking property that it completely ignores many examples on each round (by assigning zero weight), which is very helpful for the weak learning algorithm in terms of computational efficiency. To test this, we conducted experiments to compare the efficiency of AdaBoost, "NH-Boost" (an analogous boosting algorithm derived from NormalHedge) and NH-Boost.DT. All details are in Appendix G. Here we only briefly summarize the results. While the three algorithms have similar performance in terms of training and test error, NH-Boost.DT is always the fastest one in terms of running time for the same number of rounds. Moreover, the average faction of examples with zero weight is significantly higher for NH-Boost.DT than for NH-Boost (see Table 3). On one hand, this explains why NH-Boost.DT is faster (besides the reason that it does not require a numerical step). On the other hand, this also implies that NH-Boost.DT tends to achieve larger margins, since zero weight is assigned to examples with large margin. This is also confirmed by our experiments.

**Acknowledgements.** Support for this research was provided by NSF Grant #1016029. The authors thank Yoav Freund for helpful discussions and the anonymous reviewers for their comments.

## Footnotes

[1]Similar potential was also proposed in recent work [22, 25] for a different setting.

[2]"DT" stands for discrete time.

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
