[Supplementary Material]

## A   Summary of Drifting Game Variants

We study three different variants of drifting games throughout the paper, which corresponds to the Hedge setting, the multi-armed bandit problem and online convex optimization respectively. The protocols of these variants are summarized below.

---

**DGv1**

Given: a loss function $L(s) = \mathbf{1}\{s \leq -R\}$.
For $t = 1, \ldots, T$:

   1. The player chooses a distribution $\mathbf{p}_t$ over $N$ chips.
   2. The adversary decides the movement of each chip $z_{t,i} \in [-1, 1]$ subject to $\mathbf{p}_t \cdot \mathbf{z}_t \geq 0$ and $|z_{t,i} - z_{t,j}| \leq 1$ for all $i$ and $j$.

The player suffers loss $\sum_{i=1}^{N} L(\sum_{t=1}^{T} z_{t,i})$.

---

**DGv2**

Given: a loss function $L(s) = \mathbf{1}\{s \leq -R\}$.
For $t = 1, \ldots, T$:

   1. The player chooses a distribution $\mathbf{p}_t$ over $N$ chips.
   2. The adversary randomly decides the movement of each chip $z_{t,i} \geq -1$ subject to $\mathbb{E}_t[z_{t,i}] \leq 1, \mathbb{E}_t[z_{t,i}^2] \leq 1/p_{t,i}$ and $\mathbb{E}_t[\mathbf{p}_t \cdot \mathbf{z}_t] \geq 0$.

The player suffers loss $\sum_{i=1}^{N} L(\sum_{t=1}^{T} z_{t,i})$.

---

**DGv3**

Given: a compact convex set $S$, a loss function $L(s) = \mathbf{1}\{s \leq -R\}$.
For $t = 1, \ldots, T$:

   1. The player chooses a density function $p_t(\mathbf{x})$ on $S$.
   2. The adversary decides a function $z_t(\mathbf{x}) : S \to [-1, 1]$ subject to $\mathbb{E}_{\mathbf{x} \sim p_t}[z_t(\mathbf{x})] \geq 0$.

The player suffers loss $\int_{\mathbf{x} \in S} L(\sum_{t=1}^{T} z_t(\mathbf{x})) d\mathbf{x}$.

---

## B   Proof of Theorem 1

*Proof.* We first show that both conversions are valid. In Algorithm 1, it is clear that $\ell_{t,i} \geq 0$. Also, $\ell_{t,i} \leq 1$ is guaranteed due to the extra restriction of DGv1. For Algorithm 2, $z_{t,i}$ lies in $B = [-1, 1]$ since $\ell_{t,i} \in [0, 1]$, and direct computation shows $\mathbf{p}_t \cdot \mathbf{z}_t = 0 \geq \beta(= 0)$ and $|z_{t,i} - z_{t,j}| = |\ell_{t,i} - \ell_{t,j}| \leq 1$ for all $i$ and $j$.

(1) For any choices of $\mathbf{z}_t$, we have

$$\sum_{i=1}^{N} L(s_{T,i}) = \sum_{i=1}^{N} L\left(\sum_{t=1}^{N} z_{t,i}\right) \leq \sum_{i=1}^{N} L\left(\sum_{t=1}^{N} (z_{t,i} - \mathbf{p}_t \cdot \mathbf{z}_t)\right),$$

where the inequality holds since $\mathbf{p}_t \cdot \mathbf{z}_t$ is required to be nonnegative and $L$ is a nonincreasing function. By Algorithm 1, $z_{t,i} - \mathbf{p}_t \cdot \mathbf{z}_t$ is equal to $\ell_{t,i} - \mathbf{p}_t \cdot \boldsymbol{\ell}_t$, leading to

$$\sum_{i=1}^{N} L(s_{T,i}) \leq \sum_{i=1}^{N} L\left(\sum_{t=1}^{N} (\ell_{t,i} - \mathbf{p}_t \cdot \boldsymbol{\ell}_t)\right) = \sum_{i=1}^{N} \mathbf{1}\left\{R \leq \sum_{t=1}^{N} (\mathbf{p}_t \cdot \boldsymbol{\ell}_t - \ell_{t,i})\right\}.$$

Since $\mathbf{R}_T^{(i+1)/N}(\mathcal{H}) < R \le \mathbf{R}_T^{i/N}(\mathcal{H})$, we must have $\sum_{t=1}^N (\mathbf{p}_t \cdot \boldsymbol{\ell}_t - \ell_{t,j}) < R$ except for the best $i$ actions, which means $\sum_{i=1}^N L(s_{T,i}) \le i$. This holds for any choices of $\mathbf{z}_t$, so $L_T(\mathcal{D}_R) \le i/N$.

(2) By Algorithm 2 and the condition $L_T(D_R) < \epsilon$, we have

$$\frac{1}{N} \sum_{i=1}^N \mathbf{1}\left\{R \le \sum_{t=1}^N (\mathbf{p}_t \cdot \boldsymbol{\ell}_t - \ell_{t,i})\right\} = \frac{1}{N} \sum_{i=1}^N L(s_{T,i}) \le L_T(\mathcal{D}_R) < \epsilon,$$

which means there are at most $\lceil N\epsilon \rceil - 1$ actions satisfying $R \le \sum_{t=1}^N (\mathbf{p}_t \cdot \boldsymbol{\ell}_t - \ell_{t,i})$, and thus $\sum_{t=1}^N (\mathbf{p}_t \cdot \boldsymbol{\ell}_t - \ell_{t,i_\epsilon}) < R$. Since this holds for any choices of $\boldsymbol{\ell}_t$, we have $\mathbf{R}_T^\epsilon(\mathcal{H}) < R$. $\qquad\square$

## C   Summary of Hedge Algorithms and Proofs of Lemma 1, Lemma 2 and Corollary 2

Table 1: Different algorithms derived from Algorithm 3, and comparisons with NormalHedge

|  | EXP | 2-norm | NormalHedge.DT | NormalHedge |
|---|---|---|---|---|
| $\Phi_T(s)$ | $e^{-\eta(s+R)}$ | $a[s]_-^2$ | $a\left(e^{[s]_-^2/3T} - 1\right)$ | N/A |
| $p_{t,i} \propto$ | $e^{-\eta s_{t-1,i}}$ | $[s_{t-1,i}-1]_-^2$ $-[s_{t-1,i}+1]_-^2$ | $e^{[s_{t-1,i}-1]_-^2/3t}$ $-e^{[s_{t-1,i}+1]_-^2/3t}$ | $-[s_{t-1,i}]_- e^{[s_{t-1,i}]_-^2/c}$ (c is s.t. $\sum_i e^{[s_{t-1,i}]_-^2/c} = Ne$) |
| $\mathbf{R}_T^\epsilon(\mathcal{H})$ | $O\left(\sqrt{T\ln\frac{1}{\epsilon}}\right)$ | $O\left(\sqrt{T/\epsilon}\right)$ | $O\left(\sqrt{T\ln\frac{\ln T}{\epsilon}}\right)$ | $O\left(\sqrt{T\ln\frac{1}{\epsilon}} + \ln^2 N\right)$ |
| Adaptive? | No | Yes | Yes | Yes |

*Proof of Lemma 1.* It suffices to show $[s-1]_-^2 + [s+1]_-^2 \le 2[s]_-^2 + 2$. When $s \ge 0$, LHS $= [s-1]_-^2 \le 1 < 2 =$ RHS. When $s < 0$, LHS $\le (s-1)^2 + (s+1)^2 = 2s^2 + 2 =$ RHS. $\qquad\square$

*Proof of Lemma 2.* Let $F(s) = \exp\left(\frac{[s-1]_-^2}{dt}\right) + \exp\left(\frac{[s+1]_-^2}{dt}\right) - 2\exp\left(\frac{[s]_-^2}{d(t-1)}\right)$. It suffices to show

$$F(s) \le 2(b_t - b_{t-1}) = \exp\left(\frac{4}{dt}\right) - 1,$$

which is clearly true for the following 3 cases:

$$F(s) = \begin{cases} 0 & \text{if } s > 1; \\ \exp\left(\frac{(s-1)^2}{dt}\right) - 1 < \exp\left(\frac{1}{dt}\right) - 1 & \text{if } 0 < s \le 1; \\ \exp\left(\frac{(s-1)^2}{dt}\right) + 1 - 2\exp\left(\frac{s^2}{d(t-1)}\right) < \exp\left(\frac{4}{dt}\right) - 1 & \text{if } -1 < s \le 0. \end{cases}$$

For the last case $s \le -1$, if we can show that $F(s)$ is increasing in this region, then the lemma follows. Below, we show this by proving $F'(s)$ is nonnegative when $s \le -1$.

Let $h(s,c) = \frac{\partial \exp(s^2/c)}{\partial s} = \frac{2s}{c}\exp\left(\frac{s^2}{c}\right)$. $F'(s)$ can now be written as

$$F'(s) = h(s-1,c) + h(s+1,c) - 2h(s,c) + 2(h(s,c) - h(s,c')),$$

where $c = dt$ and $c' = d(t-1)$. Next we apply (one-dimensional) Taylor expansion to $h(s-1,c)$ and $h(s+1,c)$ around $s$, and $h(s,c')$ around $c$, leading to

$$F'(s) = \sum_{k=1}^\infty \frac{(-1)^k}{k!}\frac{\partial^k h(s,c)}{\partial s^k} + \sum_{k=1}^\infty \frac{1}{k!}\frac{\partial^k h(s,c)}{\partial s^k} - 2\sum_{k=1}^\infty \frac{(c'-c)^k}{k!}\frac{\partial^k h(s,c)}{\partial c^k}$$

$$= 2\sum_{k=1}^\infty \left(\frac{1}{(2k)!}\frac{\partial^{2k} h(s,c)}{\partial s^{2k}} - \frac{(-d)^k}{k!}\frac{\partial^k h(s,c)}{\partial c^k}\right).$$

Direct computation (see Lemma 3 below) shows that $\frac{\partial^k h(s,c)}{\partial c^k}$ and $\frac{\partial^{2k} h(s,c)}{\partial s^{2k}}$ share exact same forms only with different constants:

$$\frac{\partial^k h(s,c)}{\partial c^k} = \exp\left(\frac{s^2}{c}\right)\sum_{j=0}^{k}(-1)^k\alpha_{k,j}\cdot\frac{s^{2j+1}}{c^{k+j+1}},$$

$$\frac{\partial^{2k} h(s,c)}{\partial s^{2k}} = \exp\left(\frac{s^2}{c}\right)\sum_{j=0}^{k}\beta_{k,j}\cdot\frac{s^{2j+1}}{c^{k+j+1}},\tag{4}$$

where $\alpha_{k,j}$ and $\beta_{k,j}$ are recursively defined as:

$$\alpha_{k+1,j} = \alpha_{k,j-1} + (k+j+1)\alpha_{k,j},$$

$$\beta_{k+1,j} = 4\beta_{k,j-1} + (8j+6)\beta_{k,j} + (2j+3)(2j+2)\beta_{k,j+1},\tag{5}$$

with initial values $\alpha_{0,0} = \beta_{0,0} = 2$ (when $j \notin \{0,\ldots,k\}$, $\alpha_{k,j}$ and $\beta_{k,j}$ are all defined to be 0). Therefore, $F'(s)$ can be further simplified as

$$F'(s) = 2\exp\left(\frac{s^2}{c}\right)\sum_{k=1}^{\infty}\sum_{j=0}^{k}\frac{s^{2j+1}}{c^{k+j+1}}\left(\frac{\beta_{k,j}}{(2k)!} - \frac{d^k\alpha_{k,j}}{k!}\right).$$

Since $s$ is negative, it suffices to show that $\frac{\beta_{k,j}}{(2k)!} \leq \frac{d^k\alpha_{k,j}}{k!}$ holds for all $k$ and $j$, which turns out to be true as long as $d \geq 3$, as shown by induction in the technical lemma 4 below. To sum up, $\Phi_t(s-1) + \Phi_t(s+1) \leq 2\Phi_{t-1}(s)$ for all $s \in \mathbb{R}$ and $t = 2,\ldots,T$.

Finally, we need to show that Eq. (2) still holds. The inequality we just proved above implies $\sum_i \Phi_t(s_{t,i}) \leq \sum_i \Phi_{t-1}(s_{t-1,i})$ for $t = 2,\ldots,T$, as shown in Theorem 2. Thus the only thing we need to show here is the case when $t = 1$. Note that $\Phi_1(s-1) + \Phi_1(s+1) \leq 2\Phi_0(s)$ does not hold for all $s$ apparently. However, in order to prove $\sum_i \Phi_1(s_{1,i}) \leq \sum_i \Phi_0(s_{0,i})$, we in fact only need a much weaker statement: $\Phi_1(-1) + \Phi_1(1) \leq 2\Phi_0(0)$ since $s_{0,i} \equiv 0$. This is equivalent to $\exp(1/d) - 1 \leq \exp(4/d) - 1$, which is true trivially. $\qquad\square$

**Lemma 3.** *Let* $h(s,c) = \frac{2s}{c}\exp\left(\frac{s^2}{c}\right)$. *The partial derivatives of* $h(s,c)$ *satisfy Eq. (4) and (5).*

*Proof.* The base case holds trivially. Assume Eq. (4) holds for a fixed $k$. Then we have

$$\frac{\partial^{k+1} h(s,c)}{\partial c^{k+1}} = \exp\left(\frac{s^2}{c}\right)\sum_{j=0}^{k}(-1)^k\alpha_{k,j}\cdot\left(-\frac{s^2}{c^2}\frac{s^{2j+1}}{c^{k+j+1}} - (k+j+1)\frac{s^{2j+1}}{c^{k+j+2}}\right)$$

$$= \exp\left(\frac{s^2}{c}\right)\sum_{j=0}^{k}(-1)^{k+1}\alpha_{k,j}\cdot\left(\frac{s^{2(j+1)+1}}{c^{(k+1)+(j+1)+1}} + (k+j+1)\frac{s^{2j+1}}{c^{(k+1)+j+1}}\right)$$

$$= \exp\left(\frac{s^2}{c}\right)\sum_{j=0}^{k+1}(-1)^{k+1}\left(\alpha_{k,j-1} + (k+j+1)\alpha_{k,j}\right)\cdot\frac{s^{2j+1}}{c^{(k+1)+j+1}}$$

$$= \exp\left(\frac{s^2}{c}\right)\sum_{j=0}^{k+1}(-1)^{k+1}\alpha_{k+1,j}\cdot\frac{s^{2j+1}}{c^{(k+1)+j+1}},$$

and

$$\frac{\partial^{2(k+1)} h(s,c)}{\partial s^{2(k+1)}} = \partial\left[\exp\left(\frac{s^2}{c}\right)\sum_{j=0}^{k}\beta_{k,j}\cdot\left(\frac{2s^{2j+2}}{c^{k+j+2}} + (2j+1)\frac{s^{2j}}{c^{k+j+1}}\right)\right]\Big/\partial s$$

$$= \exp\left(\frac{s^2}{c}\right)\sum_{j=0}^{k}\beta_{k,j}\cdot\left(\frac{4s^{2j+3}}{c^{k+j+3}} + (8j+6)\frac{s^{2j+1}}{c^{k+j+2}} + (2j+1)2j\frac{s^{2j-1}}{c^{k+j+1}}\right)$$

$$= \exp\left(\frac{s^2}{c}\right)\sum_{j=0}^{k+1}(4\beta_{k,j-1} + (8j+6)\beta_{k,j} + (2j+3)(2j+2)\beta_{k,j+1})\cdot\frac{s^{2j+1}}{c^{k+j+2}}$$

$$= \exp\left(\frac{s^2}{c}\right) \sum_{j=0}^{k+1} \beta_{k+1,j} \cdot \frac{s^{2j+1}}{c^{k+j+2}},$$

concluding the proof. □

**Lemma 4.** *Let $\alpha_{k,j}$ and $\beta_{k,j}$ be defined as in Eq. (5). Then $\frac{\beta_{k,j}}{(2k)!} \leq \frac{d^k \alpha_{k,j}}{k!}$ holds for all $k \geq 0$ and $j \in \{0, \ldots, k\}$ when $d \geq 3$.*

*Proof.* We prove the lemma by induction on $k$. The base case $k = 0$ is trivial. Assume $\frac{\beta_{k,j}}{(2k)!} \leq \frac{d^k \alpha_{k,j}}{k!}$ holds for a fixed $k$ and all $j \in \{0, \ldots, k\}$, then we have $\forall j$,

$$\frac{\beta_{k+1,j}}{(2k+2)!} = \frac{4\beta_{k,j-1} + (8j+6)\beta_{k,j} + (2j+3)(2j+2)\beta_{k,j+1}}{(2k+2)!}$$

$$\leq \frac{d^k\left(4\alpha_{k,j-1} + (8j+6)\alpha_{k,j} + (2j+3)(2j+2)\alpha_{k,j+1}\right)}{(2k+2)(2k+1)k!}.$$

We need to show that the above expression is at most $d^{k+1}\alpha_{k+1,j}/(k+1)!$, which, after arrangements, is equivalent to $2\alpha_{k,j-1} + (4j+3)\alpha_{k,j} + (2j+3)(j+1)\alpha_{k,j+1} \leq d(2k+1)\alpha_{k+1,j}$. We will prove this by another induction on $k$. Then the lemma follows.

The base case ($k = 0$) is simplified to $6 \leq 2d$, which is true by our assumption $d \geq 3$. Assume the inequality holds for a fixed $k$, then by the definition of $\alpha_{k,j}$, one has

$$2\alpha_{k+1,j-1} + (4j+3)\alpha_{k+1,j} + (2j+3)(j+1)\alpha_{k+1,j+1}$$
$$= (2\alpha_{k,j-2} + (4j+3)\alpha_{k,j-1} + (2j+3)(j+1)\alpha_{k,j}) +$$
$$\quad (2(k+j)\alpha_{k,j-1} + (4j+3)(k+j+1)\alpha_{k,j} + (2j+3)(j+1)(k+j+2)\alpha_{k,j+1})$$
$$= (2\alpha_{k,j-2} + (4j-1)\alpha_{k,j-1} + (2j+1)j\alpha_{k,j}) +$$
$$\quad (k+j+2)\left(2\alpha_{k,j-1} + (4j+3)\alpha_{k,j} + (2j+3)(j+1)\alpha_{k,j+1}\right)$$
$$\leq d(2k+1)(\alpha_{k+1,j-1} + (k+j+2)\alpha_{k+1,j}) \qquad \text{(by induction)}$$
$$= d(2k+1)\alpha_{k+2,j}$$
$$\leq d(2k+3)\alpha_{k+2,j},$$

completing the induction. □

*Proof of Corollary 2.* Recall that $\Phi_T(s) \geq \mathbf{1}\left\{s \leq -\sqrt{dT\ln\left(\frac{1}{a}+1\right)}\right\}$. So by setting $\Phi_0(0) = a(1-b_0) < \epsilon$ and applying Theorem 2, we arrive at

$$\mathbf{R}_T^\epsilon(\mathcal{H}) \leq \sqrt{dT\ln\left(\frac{1-b_0}{\epsilon}+1\right)}.$$

It suffices to upper bound $1 - b_0$, which, by definition, is $\frac{1}{2}\sum_{t=1}^T \left(\exp\left(\frac{4}{dt}\right) - 1\right)$. Since $e^x - 1 \leq \frac{e^c-1}{c}x$ for any $x \in [0, c]$, we have

$$\sum_{t=1}^T \left(\exp\left(\frac{4}{dt}\right) - 1\right) \leq (e^{4/d} - 1)\sum_{t=1}^T \frac{1}{t} \leq (e^{4/d} - 1)(\ln T + 1).$$

Plugging $d = 3$ gives the corollary. □

## D A General MAB Algorithm and Regret Bounds

**Input**: A convex, nonincreasing, nonnegative function $\Phi_T(s) \in \mathbb{C}^2$, with nonincreasing second derivative.

**for** $t = T$ **down to** $1$ **do**
&emsp;Find a convex function $\Phi_{t-1}(s)$ s.t. the conditions of Theorem 4 hold.

Set: $\mathbf{s}_0 = \mathbf{0}$.

**for** $t = 1$ **to** $T$ **do**
&emsp;Set: $p_{t,i} \propto \Phi_t(s_{t-1,i} - 1) - \Phi_t(s_{t-1,i} + 1)$.
&emsp;Draw $i_t \sim \mathbf{p}_t$ and receive loss $\ell_{t,i_t}$.
&emsp;Set: $z_{t,i} = \mathbf{1}\{i = i_t\} \cdot \ell_{t,i_t}/p_{t,i_t} - \ell_{t,i_t}, \; \forall i$.
&emsp;Set: $\mathbf{s}_t = \mathbf{s}_{t-1} + \mathbf{z}_t$.

**Algorithm 4:** A General MAB Algorithm

**Theorem 4.** *Suppose $\Phi_t(s)$ is convex, twice continuously differentiable (i.e. $\Phi_t(s) \in \mathbb{C}^2$), have nonincreasing second derivative, and satisfies:*

$$\left(\tfrac{1}{2} + N\alpha_t\right)\Phi_t(s-1) + \left(\tfrac{1}{2} - N\alpha_t\right)\Phi_t(s+1) \le \Phi_{t-1}(s), \forall s \in \mathbb{R} \tag{6}$$

*where $\alpha_t = \frac{1}{2}\max_s \frac{\Phi_t''(s-1)}{\Phi_t(s-1) - \Phi_t(s+1)}$. If the player's strategy is such that $p_{t,i} \propto \Phi_t(s_{t-1,i} - 1) - \Phi_t(s_{t-1,i} + 1)$, then Eq. (2) holds in expectation.*

*Proof of Theorem 4.* As discussed before, the main difficulty here is the unboundedness of $z_{t,i}$. However, the expectation of $z_{t,i}$ is still in $[-1, 1]$ as in DGv1. To exploit this fact, we apply Taylor's theorem to $\Phi_t(s_{t-1,i} + z_{t,i})$ to the second order term:

$$
\begin{aligned}
\Phi_t(s_{t,i}) &= \Phi_t(s_{t-1,i} + z_{t,i}) \\
&= \Phi_t(s_{t-1,i}) + \Phi_t'(s_{t-1,i})z_{t,i} + \tfrac{1}{2}\Phi_t''(\xi_{t,i})z_{t,i}^2 \\
&\le \Phi_t(s_{t-1,i}) + \Phi_t'(s_{t-1,i})z_{t,i} + \tfrac{1}{2}\Phi_t''(s_{t-1,i} - 1)z_{t,i}^2,
\end{aligned}
$$

where $\xi_{t,i}$ is between $s_{t-1,i} + z_{t,i}$ and $s_{t-1,i}$, and the inequality holds because $\Phi_t''(s)$ is nonincreasing and $z_{t,i} \ge -1$ by assumption. Now taking expectation on both sides with respect to the randomness of $z_{t,i}$, using the convexity of $\Phi_t(s)$, and plugging the assumption $\mathbb{E}_t[z_{t,i}^2] \le 1/p_{t,i}$ give:

$$
\begin{aligned}
\mathbb{E}_t[\Phi_t(s_{t,i})] &\le \Phi_t(s_{t-1,i}) + \Phi_t'(s_{t-1,i})\mathbb{E}_t[z_{t,i}] + \tfrac{1}{2}\Phi_t''(s_{t-1,i} - 1)\mathbb{E}_t[z_{t,i}^2] \\
&\le \Phi_t\left(s_{t-1,i} + \mathbb{E}_t[z_{t,i}]\right) + \tfrac{1}{2}\Phi_t''(s_{t-1,i} - 1)/p_{t,i}.
\end{aligned}
$$

Let $w_{t,i} = \frac{1}{2}\left(\Phi_t(s_{t-1,i} - 1) - \Phi_t(s_{t-1,i} + 1)\right)$. Further plugging $\mathbf{p}_{t,i} \propto w_{t,i}$ and summing over all $i$, we arrive at

$$
\begin{aligned}
\sum_{i=1}^N \mathbb{E}_t[\Phi_t(s_{t,i})] &\le \sum_{i=1}^N \left( \Phi_t\left(s_{t-1,i} + \mathbb{E}_t[z_{t,i}]\right) + \frac{\Phi_t''(s_{t-1,i} - 1)}{2w_{t,i}} \cdot \sum_{i=1}^N w_{t,i} \right) \\
&\le \sum_{i=1}^N \left( \Phi_t\left(s_{t-1,i} + \mathbb{E}_t[z_{t,i}]\right) + 2\alpha_t \sum_{i=1}^N w_{t,i} \right) \qquad \text{(by the defintion of } \alpha_t) \\
&= \sum_{i=1}^N \left( \Phi_t\left(s_{t-1,i} + \mathbb{E}_t[z_{t,i}]\right) + 2N\alpha_t w_{t,i} \right).
\end{aligned}
$$

Since $\mathbb{E}_t[\mathbf{p}_t \cdot \mathbf{z}_t] \ge 0$ implies $\sum_{i=1}^N w_{t,i}\mathbb{E}_t[z_{t,i}] \ge 0$, we thus have

$$
\begin{aligned}
\sum_{i=1}^N \mathbb{E}_t[\Phi_t(s_{t,i})] &\le \sum_{i=1}^N \left( \Phi_t\left(s_{t-1,i} + \mathbb{E}_t[z_{t,i}]\right) + w_{t,i}\mathbb{E}_t[z_{t,i}] + 2N\alpha_t w_{t,i} \right) \\
&\le \sum_{i=1}^N \left( \max_{z \in [-1,+1]} \left(\Phi_t\left(s_{t-1,i} + z\right) + w_{t,i}z\right) + 2N\alpha_t w_{t,i} \right)
\end{aligned}
$$

$$= \sum_{i=1}^{N} \left( \max_{z \in \{-1,+1\}} \left( \Phi_t \left( s_{t-1,i} + z \right) + w_{t,i} z \right) + 2N\alpha_t w_{t,i} \right)$$

$$\text{(by the convexity of } \Phi_t(s))$$

$$= \sum_{i=1}^{N} \left( \left( \tfrac{1}{2} + N\alpha_t \right) \Phi_t(s_{t-1,i} - 1) + \left( \tfrac{1}{2} - N\alpha_t \right) \Phi_t(s_{t-1,i} + 1) \right)$$

$$\leq \sum_{i=1}^{N} \Phi_{t-1}(s_{t-1,i}). \qquad \text{(by assumption)}$$

The theorem follows by taking expectation on both sides with respect to the past (i.e. the randomness of $\mathbf{z}_1, \ldots, \mathbf{z}_{t-1}$). $\qquad \square$

**Theorem 5.** *For Algorithm 4, if $R$ and $\epsilon$ are such that $\Phi_0(0) < \epsilon$ and $\Phi_T(s) \geq \mathbf{1}\{s \leq -R\}$ for all $s \in \mathbb{R}$, then $\mathbb{E}[\sum_{t=1}^{T} \ell_{t,i_t} - \sum_{t=1}^{T} \ell_{t,i_\epsilon}] < R$ for any non-oblivious adversary. Moreover, using $\Phi_T(s) = \exp(-\eta(s + R))$ (and let Eq. (6) hold with equality) gives exactly the EXP3 algorithm with regret $O(\sqrt{TN \ln(1/\epsilon)})$.*

*Proof of Theorem 5.* We first show that Algorithm 4 converts the multi-armed bandit problem to a valid instance of DGv2. It suffices to prove that $z_{t,i} = \mathbf{1}\{i = i_t\} \cdot \ell_{t,i_t}/p_{t,i_t} - \ell_{t,i_t}$ satisfies all conditions defined in DGv2, as shown below ($z_{t,i} \geq -1$ is trivial):

$$\mathbb{E}_t[z_{t,i}] = \ell_{t,i} - \mathbf{p}_t \cdot \boldsymbol{\ell}_t \leq 1,$$

$$\mathbb{E}_t[z_{t,i}^2] = p_{t,i} \left( \frac{\ell_{t,i}}{p_{t,i}} - \ell_{t,i} \right)^2 + \sum_{j \neq i} p_{t,j} \ell_{t,j}^2 \leq p_{t,i} \left( \frac{1}{p_{t,i}} - 1 \right)^2 + \sum_{j \neq i} p_{t,j} = \frac{1}{p_{t,i}} - 1 \leq \frac{1}{p_{t,i}},$$

$$\mathbb{E}_t[\mathbf{p}_t \cdot \mathbf{z}_t] = \mathbb{E}_t \left[ \ell_{t,i_t} - \sum_{j=1}^{N} p_{t,j} \ell_{t,i_t} \right] = 0.$$

Therefore, we can apply Theorem 4 directly, arriving at:

$$\frac{1}{N} \sum_{i=1}^{N} \mathbb{E}[\Phi_T(s_{T,i})] \leq \cdots \leq \frac{1}{N} \sum_{i=1}^{N} \mathbb{E}[\Phi_0(s_{0,i})] = \Phi_0(0) \leq \epsilon.$$

On the other hand, by applying Jensen' inequality, we have

$$\mathbb{E}[\Phi_T(s_{T,i})] \geq \Phi_T(\mathbb{E}[s_{T,i}]) \geq \mathbf{1}\{\mathbb{E}[s_{T,i}] \leq -R\}.$$

Note that $\mathbb{E}[s_{T,i}]$ is equal to $\mathbb{E}\left[ \sum_{t=1}^{T} (\ell_{t,i} - \ell_{t,i_t}) \right]$. We thus know

$$\frac{1}{N} \sum_{i=1}^{N} \mathbf{1} \left\{ \mathbb{E} \left[ \sum_{t=1}^{T} (\ell_{t,i} - \ell_{t,i_t}) \right] \leq -R \right\} < \epsilon,$$

which implies $\mathbb{E}\left[ \sum_{t=1}^{T} \ell_{t,i_t} - \sum_{t=1}^{T} \ell_{t,i_\epsilon} \right] < R$ for any non-oblivious adversary for the exact same argument used in the proof of Theorem 2.

Finally, we show how to recover EXP3 using Algorithm 4 with input $\Phi_T(s) = \exp(-\eta(s + R))$. To compute $\Phi_t(s)$ for $t < T$, we simply use Eq. (6) with equality. One can verify using induction that

$$\Phi_t(s) = \exp\left(-\eta(s + R)\right) \left( \frac{e^\eta + e^{-\eta} + N e^\eta \eta^2}{2} \right)^{T-t},$$

$$\alpha_t = \frac{1}{2} \max_s \frac{\eta^2 \Phi_t(s-1)}{\Phi_t(s-1) - \Phi_t(s+1)} = \frac{e^\eta \eta^2}{2(e^\eta - e^{-\eta})},$$

$$\Phi_t'''(s) = -\eta^3 \Phi_t(s) \leq 0.$$

The player's strategy is thus $\mathbf{p}_{t,i} \propto \exp(-\eta \sum_{\tau=1}^{t-1} \hat{\ell}_{\tau,i})$ (recall $\hat{\ell}_{t,i} = \mathbf{1}\{i = i_t\} \cdot \ell_{t,i_t}/p_{t,i_t}$ is the estimated loss), which is exactly the same as EXP3 (in fact a simplified version of the original EXP3, see for example [30]). Moreover, the regret can be computed by setting $\Phi_0(0) = \epsilon$, leading to

$$\begin{aligned}
R &= \frac{1}{\eta} \ln\left(\frac{1}{\epsilon}\right) + \frac{T}{\eta} \ln\left(\frac{e^\eta + e^{-\eta}}{2} + \frac{1}{2}Ne^\eta\eta^2\right) \\
&\leq \frac{1}{\eta} \ln\left(\frac{1}{\epsilon}\right) + \frac{T}{\eta} \ln\left(e^{\eta^2/2} + \frac{1}{2}Ne^\eta\eta^2\right) && \text{(by Hoeffding's Lemma)} \\
&\leq \frac{1}{\eta} \ln\left(\frac{1}{\epsilon}\right) + \frac{T}{\eta} \left(\frac{\eta^2}{2} + \frac{1}{2}Ne^{\eta - \frac{\eta^2}{2}}\eta^2\right) && (\ln(1+x) \leq x)
\end{aligned}$$

If $\eta \leq 1$ so that $e^{\eta - \eta^2/2} \leq \sqrt{e}$, then we have $R \leq \frac{1}{\eta}\ln(\frac{1}{\epsilon}) + T\eta\left(\frac{1}{2} + \frac{N\sqrt{e}}{2}\right)$, which is $\sqrt{2T(1 + N\sqrt{e})\ln(1/\epsilon)}$ after optimally choosing $\eta$ ($\eta \leq 1$ will be satisfied when $T$ is large enough). $\qquad\square$

## E  A General OCO Algorithm and Regret Bounds

---
**Input**: A convex, nonincreasing, nonnegative function $\Phi_T(s)$
**for** $t = T$ **down to** $1$ **do**
    Find a convex function $\Phi_{t-1}(s)$ s.t. $\forall s, \Phi_t(s-1) + \Phi_t(s+1) \leq 2\Phi_{t-1}(s)$.
Set: $s_0(x) \equiv 0$.
**for** $t = 1$ **to** $T$ **do**
    Predict $\mathbf{x}_t = \mathbb{E}_{\mathbf{x} \sim p_t}[\mathbf{x}]$ where $p_t$ is such that $p_t(\mathbf{x}) \propto \Phi_t(s_{t-1}(\mathbf{x}) - 1) - \Phi_t(s_{t-1}(\mathbf{x}) + 1)$.
    Receive loss function $f_t$ from the adversary.
    Set: $z_t(\mathbf{x}) = f_t(\mathbf{x}) - f_t(\mathbf{x}_t)$.
    Set: $s_t(\mathbf{x}) = s_{t-1}(\mathbf{x}) + z_t(\mathbf{x})$.

---
**Algorithm 5:** A General OCO Algorithm

**Definition of $\epsilon$-regret in the OCO setting**: Let $S_\epsilon \subset S$ be such that the ratio of its volume and the one of $S$ is $\epsilon$ and also $\sum_{t=1}^T f_t(\mathbf{x}') \leq \sum_{t=1}^T f_t(\mathbf{x})$ for all $\mathbf{x}' \in S_\epsilon$ and $\mathbf{x} \in S \backslash S_\epsilon$ (it is clear that such set exists). Then $\epsilon$-regret is defined as $\mathbf{R}_T^\epsilon(\mathbf{x}_{1:T}, f_{1:T}) = \sum_{t=1}^T f_t(\mathbf{x}_t) - \inf_{\mathbf{x} \in S \backslash S_\epsilon} \sum_{t=1}^T f_t(\mathbf{x})$.

**Theorem 6.** *For Algorithm 5, if $R$ is such that $\Phi_T(s) \geq \mathbf{1}\{s \leq -R\}$ and $\Phi_0(0) < \epsilon$, then we have $\mathbf{R}_T^\epsilon(\mathbf{x}_{1:T}, f_{1:T}) < R$ and $\mathbf{R}_T(\mathbf{x}_{1:T}, f_{1:T}) < R + T\epsilon^{1/d}$. Specifically, if $R = O(\sqrt{T\ln(1/\epsilon)})$, then setting $\epsilon = T^{-d}$ gives $\mathbf{R}_T(\mathbf{x}_{1:T}, f_{1:T}) = O(\sqrt{dT\ln T})$.*

*Proof of Theorem 6.* Let $w_t(\mathbf{x}) = \frac{1}{2}\left(\Phi_t(s_{t-1}(\mathbf{x}) - 1) - \Phi_t(s_{t-1}(\mathbf{x}) + 1)\right)$. Similarly to the Hedge setting, the "sum" of potentials never increases:

$$\int_{\mathbf{x} \in S} \Phi_t(s_t(\mathbf{x}))d\mathbf{x} \leq \int_{\mathbf{x} \in S} \left(\Phi_t(s_{t-1}(\mathbf{x}) + z_t(\mathbf{x})) + w_t(\mathbf{x})z_t(\mathbf{x})\right)d\mathbf{x} \leq \int_{\mathbf{x} \in S} \Phi_{t-1}(s_{t-1}(\mathbf{x}))d\mathbf{x}.$$

Here, the first inequality is due to $\mathbb{E}_{\mathbf{x} \sim p_t}[z_t(\mathbf{x})] \geq 0$, and the second inequality holds for the exact same reason as in the case for Hedge. Therefore, we have

$$\int_{\mathbf{x} \in S} \mathbf{1}\{s_T(\mathbf{x}) \leq -R\}d\mathbf{x} \leq \int_{\mathbf{x} \in S} \Phi_T(s_T(\mathbf{x}))d\mathbf{x} \leq \cdots \leq \int_{\mathbf{x} \in S} \Phi_0(0)d\mathbf{x} < \epsilon V,$$

where $V$ is the volume of $S$. Recall the construction of $S_\epsilon$. There must exist a point $\mathbf{x}' \in S_\epsilon$ such that $s_T(\mathbf{x}') > -R$, otherwise $\int_{\mathbf{x}} \mathbf{1}\{s_T(\mathbf{x}) \leq -R\}d\mathbf{x}$ would be at least $\epsilon V$. Unfolding $s_T(\mathbf{x}')$, we arrive at $\sum_t f_t(\mathbf{x}_t) - \sum_t f_t(\mathbf{x}') < R$. Using the fact $\sum_t f_t(\mathbf{x}') \leq \inf_{\mathbf{x} \in S \backslash S_\epsilon} \sum_t f_t(\mathbf{x})$ gives the bound for $\epsilon$-regret.

Next consider a shrunk version of $S$: $S_\epsilon' = \{(1 - \epsilon^{\frac{1}{d}})\mathbf{x}^* + \epsilon^{\frac{1}{d}}\mathbf{x} : \mathbf{x} \in S\}$ where $\mathbf{x}^* \in \arg\min_{\mathbf{x}} \sum_t f_t(\mathbf{x})$. Then $\int_{\mathbf{x} \in S} \mathbf{1}\{s_T(\mathbf{x}) \leq -R\}d\mathbf{x}$ is at least

$$\int_{\mathbf{x} \in S_\epsilon'} \mathbf{1}\{s_T(\mathbf{x}) \leq -R\}d\mathbf{x} = \epsilon \int_{\mathbf{x} \in S} \mathbf{1}\left\{s_T\left((1 - \epsilon^{\frac{1}{d}})\mathbf{x}^* + \epsilon^{\frac{1}{d}}\mathbf{x}\right) \leq -R\right\}d\mathbf{x},$$

which, by the convexity and the boundedness of $f_t(\mathbf{x})$, is at least

$$\epsilon \int_{\mathbf{x} \in S} \mathbf{1} \left\{ \sum_{t=1}^{T} \left( (1 - \epsilon^{\frac{1}{d}}) f_t(\mathbf{x}^*) + \epsilon^{\frac{1}{d}} f_t(\mathbf{x}) - f_t(\mathbf{x}_t) \right) \leq -R \right\} d\mathbf{x}$$

$$\geq \epsilon \int_{\mathbf{x} \in S} \mathbf{1} \left\{ \sum_{t=1}^{T} (f_t(\mathbf{x}^*) - f_t(\mathbf{x}_t)) \leq -R - T\epsilon^{\frac{1}{d}} \right\} d\mathbf{x}$$

$$= \epsilon V \cdot \mathbf{1} \left\{ \sum_{t=1}^{T} (f_t(\mathbf{x}^*) - f_t(\mathbf{x}_t)) \leq -R - T\epsilon^{\frac{1}{d}} \right\}.$$

Following the previous discussion, the expression in the last line above is strictly less than $\epsilon V \cdot$, which means that the value of the indicator function has to be 0, namely, $\mathbf{R}_T(\mathbf{x}_{1:T}, f_{1:T}) < R + T\epsilon^{1/d}$. $\qquad\square$

## F NH-Boost.DT, NH-Boost and Proof of Theorem 3

---
**Input** : Training examples $(\mathbf{x}_i, y_i) \in \mathbb{R}^d \times \{-1, +1\}, i = 1, \ldots, N$.
**Input** : A weak learning algorithm.
**Input** : Number of rounds $T$.
**Output**: A Hypothesis $H(\mathbf{x}) : \mathbb{R}^d \to \{-1, +1\}$.
Set: $\mathbf{s}_0 = \mathbf{0}$.
**for** $t = 1$ **to** $T$ **do**
    Set: $p_{t,i} \propto \exp\left([s_{t-1,i} - 1]_-^2 / 3t\right) - \exp\left([s_{t-1,i} + 1]_-^2 / 3t\right), \forall i$.
    Invoke the weak learning algorithm to get $h_t$ with edge $\gamma_t = \frac{1}{2} \sum_i p_{t,i} y_i h_t(\mathbf{x}_i)$.
    Set: $s_{t,i} = s_{t-1,i} + \frac{1}{2} y_i h_t(\mathbf{x}_i) - \gamma_t, \forall i$.
Set: $H(\mathbf{x}) = \text{sign}(\sum_{t=1}^{T} h_t(\mathbf{x}))$.

---
**Algorithm 6:** NH-Boost.DT

---
**Input** : Training examples $(\mathbf{x}_i, y_i) \in \mathbb{R}^d \times \{-1, +1\}, i = 1, \ldots, N$.
**Input** : A weak learning algorithm.
**Input** : Number of rounds $T$.
**Output**: A Hypothesis $H(\mathbf{x}) : \mathbb{R}^d \to \{-1, +1\}$.
Set: $\mathbf{s}_0 = \mathbf{0}$.
**for** $t = 1$ **to** $T$ **do**
    **if** $t = 1$ **then**
        Set: $\mathbf{p}_1$ to be a uniform distribution.
    **else**
        Find: $c$ such that $\sum_{i=1}^{N} \exp\left([s_{t-1,i}]_-^2 / c\right) = Ne$.
        Set: $p_{t,i} \propto -[s_{t-1,i}]_- \exp\left([s_{t-1,i}]_-^2 / c\right), \forall i$.
    Invoke the weak learning algorithm to get $h_t$ with edge $\gamma_t = \frac{1}{2} \sum_i p_{t,i} y_i h_t(\mathbf{x}_i)$.
    Set: $s_{t,i} = s_{t-1,i} + \frac{1}{2} y_i h_t(\mathbf{x}_i) - \gamma_t, \forall i$.
Set: $H(\mathbf{x}) = \text{sign}(\sum_{t=1}^{T} h_t(\mathbf{x}))$.

---
**Algorithm 7:** NH-Boost

In the boosting setting for binary classification, we are given a set of training examples $(\mathbf{x}_i, y_i)_{i=1,\ldots,N}$ where $\mathbf{x}_i \in \mathbb{R}^d$ is an example and $y_i \in \{-1, +1\}$ is its label. A boosting algorithm proceeds for $T$ rounds. On each round, a distribution $\mathbf{p}_t$ over the examples is computed and fed into a weak learning algorithm which returns a "weak" hypothesis $h_t : \mathbb{R}^d \to \{-1, +1\}$ with a guaranteed small edge, that is, $\gamma_t = \frac{1}{2} \sum_i p_{t,i} y_i h_t(\mathbf{x}_i) \geq \gamma > 0$. At the end, a linear combination of all $h_t$ is computed as the final "strong" hypothesis which is expected to have low training error and potentially low generalization error.

The conversion of a Hedge algorithm into a boosting algorithm is to treat each example as an "action" and set $\ell_{t,i} = \mathbf{1}\{h_t(\mathbf{x}_i) = y_i\}$ so that the booster tends to increase weights for those "hard"

examples. The final hypothesis is a simple majority vote of all $h_t$, that is, $H(\mathbf{x}) = \text{sign}(\sum_t h_t(\mathbf{x}))$ where $\text{sign}(x)$ is the sign function that outputs 1 if $x$ is positive, and $-1$ otherwise. The *margin* of example $\mathbf{x}_i$ is defined as $\frac{1}{T}\sum_{t=1}^{T} y_i h_t(\mathbf{x}_i)$, that is, the difference between the fractions of correct hypotheses and incorrect hypotheses on this example. The boosting algorithms derived from NormalHedge.DT and NormalHedge in this way are given in Algorithm 6 and 7.

*Proof the Theorem 3.* Let $(\tilde{\mathbf{x}}_i, \tilde{y}_i)_{i=1,\ldots,N}$ be a permutation of the training examples such that their margins are sorted from smallest to largest: $\sum_t \tilde{y}_1 h_t(\tilde{\mathbf{x}}_1) \leq \cdots \leq \sum_t \tilde{y}_N h_t(\tilde{\mathbf{x}}_N)$, which also implies $\sum_t \mathbf{1}\{h_t(\tilde{\mathbf{x}}_1) = \tilde{y}_1\} \leq \cdots \leq \sum_t \mathbf{1}\{h_t(\tilde{\mathbf{x}}_N) = \tilde{y}_N\}$. Recall that NormalHedge.DT is essentially playing a Hedge game using NormalHedge.DT with loss $\ell_{t,i} = \mathbf{1}\{h_t(\mathbf{x}_i) = y_i\}$. Therefore, the $\epsilon$-regret bound for the Hedge setting together with the assumption on the weak learning algorithm implies: $\forall j \in \{1,\ldots,N\}$,

$$\frac{1}{2} + \gamma \leq \frac{1}{T}\sum_{t=1}^{T}\sum_{i=1}^{N} \mathbf{p}_{t,i}\mathbf{1}\{h_t(\mathbf{x}_i) = y_i\} \leq \frac{1}{T}\sum_{t=1}^{T}\mathbf{1}\{h_t(\tilde{\mathbf{x}}_j) = \tilde{y}_j\} + \frac{\mathbf{R}_T^{j/N}}{T}, \qquad (7)$$

where $\mathbf{R}_T^{j/N} = \tilde{O}(\sqrt{3T\ln(N/j)})$ is the $j/N$-regret bound for NormalHedge.DT. So if $j$ is such that $\gamma > \mathbf{R}_T^{j/N}/T$, we have $\frac{1}{T}\sum_{t=1}^{T}\mathbf{1}\{h_t(\tilde{\mathbf{x}}_j) = \tilde{y}_j\} > \frac{1}{2}$, which is saying that example $(\tilde{\mathbf{x}}_j, \tilde{y}_j)$ will eventually be classified correctly by $H(\mathbf{x})$ due to the fact that $H(\mathbf{x})$ is taking a majority vote of all $h_t$. This is in fact true for all examples $(\tilde{\mathbf{x}}_i, \tilde{y}_i)$ such that $i \geq j$ and thus the training error rate will be at most $(j-1)/N$, which is of order $\tilde{O}(\exp(-\frac{1}{3}T\gamma^2))$.

For the margin bound, by plugging $\mathbf{1}\{h_t(\tilde{\mathbf{x}}_j) = \tilde{y}_j\} = (\tilde{y}_j h_t(\tilde{\mathbf{x}}_j) + 1)/2$, we rewrite Eq. (7) as:

$$2\left(\gamma - \frac{\mathbf{R}_T^{j/N}}{T}\right) \leq \frac{1}{T}\sum_{t=1}^{T}\tilde{y}_j h_t(\tilde{\mathbf{x}}_j).$$

Therefore, if $j$ is such that $\theta < 2(\gamma - \mathbf{R}_T^{j/N}/T)$, then the fraction of examples with margin at most $\theta$ is again at most $(j-1)/N$, which is of order $\tilde{O}(\exp(-\frac{1}{3}T(\theta - 2\gamma)^2))$. $\qquad \square$

## G   Experiments in a Boosting Setting

We conducted experiments to compare the performance of three boosting algorithms for binary classification: AdaBoost [14], NH-Boost (Algorithm 7) and NH-Boost.DT (Algorithm 6), using a set of benchmark data available from the UCI repository[3] and LIBSVM datasets[4]. Some datasets are preprocessed according to [27]. The number of features, training examples and test examples can be found in Table 2.

All features are binary. The weak learning algorithm is a simple (exhaustive) decision stump (see for instance [29]). On each round, the weak learning algorithm enumerates all features, and for each feature computes the weighted error of the corresponding stump on the weighted training examples. Therefore, if the number of examples with zero weight is relatively large, then the weak learning algorithm would be faster in computing the weighted error and thus faster in finding the best feature.

All boosting algorithms are run for two hundred rounds. The results are summarized in Table 3, with bold entries being the best ones among the three (AB, NB and NBDT stand for AdaBoost, NH-Boost and NH-Boost.DT respectively). As we can see, in terms of training error and test error, all three algorithms have similar performance. However, our NH-Boost.DT algorithm is always the fastest one. The average fraction of examples with zero weights for NH-Boost.DT is significantly higher than the one for NH-Boost (note that AdaBoost does not assign zero weight at all). We plot the change of this fraction over rounds in Figure 1 (using three datasets). As both algorithms proceed, they tend to ignore more and more examples on each round, but NH-Boost.DT consistently ignores more examples than NH-Boost.

Since $s_{t,i}$ is positively correlated to the margin of example $i$ ($\frac{1}{t}\sum_{\tau=1}^{t} y_i h_\tau(\mathbf{x}_i)$) and large $s_{t,i}$ leads to zero weight, the above phenomenon in fact implies that the examples' margins should be larger for

NH-Boost.DT than for NH-Boost. This is confirmed by Figure 2, where we plot the final cumulative margins on three datasets (i.e. each point represents the fraction of examples with at most some fixed margin). One can see that the lines for NH-Boost.DT are below the ones for NH-Boost (and even AdaBoost) for most time, meaning that NH-Boost.DT achieves larger margins in general. This could explain NH-Boost.DT's better test error on some datasets.

Table 2: Description of datasets

| Data | #feature | #training | #test |
|---|---|---|---|
| a9a | 123 | 32,561 | 16,281 |
| census | 131 | 1,000 | 1,000 |
| ocr49 | 403 | 1,000 | 1,000 |
| splice | 240 | 500 | 500 |
| w8a | 300 | 49,749 | 14,951 |

Table 3: Experiment results

| | Time (s) | | | Zeros (%) | | Training Error (%) | | | Test Error (%) | | |
|---|---|---|---|---|---|---|---|---|---|---|---|
| Data | AB | NB | NBDT | NB | NBDT | AB | NB | NBDT | AB | NB | NBDT |
| a9a | 57.5 | 72.5 | **46.2** | 1.1 | **22.1** | **15.4** | 15.8 | 15.5 | **15.0** | 15.6 | 15.2 |
| census | 1.7 | 2.2 | **1.4** | 2.2 | **19.2** | 15.6 | 17.0 | **15.4** | 18.7 | 18.6 | **18.3** |
| ocr49 | 5.1 | 4.7 | **3.0** | 17.1 | **42.0** | **1.7** | **1.7** | 2.4 | **5.5** | 5.9 | 5.8 |
| splice | 1.6 | 1.5 | **0.9** | 22.2 | **45.1** | **0.0** | **0.0** | 0.4 | 9.4 | 8.6 | **8.2** |
| w8a | 237.6 | 244.7 | **170.7** | 3.0 | **29.3** | 2.6 | **2.2** | 2.4 | 2.7 | **2.3** | 2.6 |

(a) census       (b) splice       (c) w8a

Figure 1: Comparison of fraction of zero weights

(a) census       (b) splice       (c) w8a

Figure 2: Comparison of cumulative margins

## Footnotes

[3]http://archive.ics.uci.edu/ml/

[4]http://www.csie.ntu.edu.tw/~cjlin/libsvmtools/datasets/