[Reviews · NeurIPS 2014]

Submitted by Assigned_Reviewer_34

The authors reduce online learning problems (linear regret/bandit, online convex optimization, boosting) into another framework known as "drifting games". It is quite related to a similar idea of Rakhlin, Shamir & Sridharan and can be seen as the decrease of some potential mappings (as it is really usual know, see the textbook of Cesa-Bianchi & Lugosi).

Those reductions give another interpretation to famous online algorithms (exponential weights, squared potential, etc) and allow to recover their guarantees with new proofs. This is interesting from a theoretical point of view, but I do not see the impact it could have.

If this approach could lead to a new class of algorithms that actually improve the state of the art (that could not be obtained by the associated potential minimization techniques), then undoubtedly this paper would be really interesting. For the moment, it is regretfully not very convincing.
Summary: The reduction of online learning problems into another framework is interesting, but it is not really original, it does not really provide new results (but allows to recover known ones).

Submitted by Assigned_Reviewer_38

The paper derives several classical online learning algorithms using the drifting-games framework beyond previous known work that had established connections between the two areas. It first introduces a specific drifting game DGv1 and establishes algorithmic equivalency with the Hedge problem in terms of a relaxed notion of external regret known as \eps-regret. Then it relaxes conditions on the drifting game, and constructs a general framework for solving these games based on minimax analysis and potential functions. Using this recipe, the authors produce algorithms for the Hedge problem dependent on the specific potential function chosen, some of which are parameter independent and adaptive to the final time T. The paper then extends this idea to derive similar results for the multi-armed bandit problem by introducing a variance constraint in the drifting game and finding a suitable potential function. In an analogous fashion, the authors derive an algorithm in online convex optimization by changing the discrete drifting game into a continuous variant. Lastly, the paper uses the drifting game framework to derive a boosting algorithm, which due to the use of \eps-regret can ignore many examples per round and is shown empirically to be more computationally efficient than other algorithms such as Adaboost.

Quality and Clarity
The paper is for the most part well-written and builds upon itself nicely by first introducing the general mechanism and then producing specific examples. One potential complaint might be that while the proofs and results all appear correct, the authors do not really take the time to explain the intuitive relationship and connection between drifting games and online learning.

Originality and Significance
While the paper largely reproves known algorithms and regret bounds, finding new ways to understand state-of-the-art online learning algorithms is an important task, and the authors accomplish this in a detailed and extensive fashion. Moreover, the authors also produce original algorithms in the same spirit as in their derivation of the known algorithms, and they accompany this work with empirical results.
Summary: The paper provides a fresh and comprehensive treatment for understanding online learning algorithms through the drifting-games framework. It could appeal more to the reader’s intuition, but the content and quality is solid.

Submitted by Assigned_Reviewer_42

This paper proposes a new scheme for constructing and analyzing algorithms for online learning based on the equivalence of a broad class of online optimization games and the class of so-called drifting games. This newly found equivalence not only allows the authors to provide an alternative derivation for the well-known exponential-weights algorithm for prediction with expert advice and online convex optimization, but also leads to a new parameter-free learning algorithm called AbnormalHedge (a variant of NormalHedge originally proposed by Chaudhuri, Freund and Hsu, 2009). The authors also use their technique to prove strong guarantees on the so-called \epsilon-regret of the resulting algorithms, defined as the regret against the best \epsilon-quantile of the pool of experts/decision space. Finally, the authors apply AbnormalHedge in a boosting setting and highlight several useful properties of the resulting boosting algorithm.

Originality & Significance
--------------------------
The results of the paper clearly constitute a substantial step forward in understanding potential-based algorithms for online learning. While this is not the first work to draw connections between drifting games and online learning, the general recipe of constructing near-optimal online learning algorithms by relaxing the minimax optimal strategy is novel. These results tie in nicely with the recent work of Rakhlin, Shamir and Sridharan (2012), who proposed a recipe for constructing online learning algorithms that work by relaxing the minimax strategy for the online learning game itself. The power of the proposed approach is clearly demonstrated by the simple and straightforward analysis provided for AbnormalHedge, which is especially elegant when compared to the complicated original analysis of NormalHedge. To the best of my knowledge, this is also the first time that the continuous exponential-weights algorithm is derived via a (relaxed) minimax analysis. The extension of the results to the multi-armed bandit setting -- while providing no spectacular new results -- can also serve as basis for interesting future work.

Clarity & Quality
-----------------
The writing is superb, there is really not much to fix except a handful of typos (see below). The technical quality is also very good for the most part, the derivations are clean and seem correct. The only exception is Section 3.4, where the authors claim that their approach can yield high-probability regret guarantees without relying on concentration-of-measure results. Although I can see how this result follows from a modification of Theorem 2, it would be probably better to provide this proof at least in the appendix for completeness.

Detailed comments
-----------------
212: \Phi -> \Phi_t in the expression of the optimal value of w. Same typo on line 229.
212: "0-1 loss function" -> "the 0-1 loss function"
234-235: This discussion is a bit difficult to follow. How does one "solve \Phi_0(0)<\epsilon" and how does such a procedure "give R > \underline{R}"?
289: The logical expression in the indicator is a bit difficult to understand: how is a constant compared against an asymptotic expression? This should be rephrased.
309: This result probably follows from an optimized setting of "a", which should be given in the text.
381: The EWOO algorithm of Hazan, Agarwal and Kale (2007) precedes the work of Narayanan and Rakhlin on continuous EXP, and this is also probably a better reference for the trick used for proving Thm 6 than [7].
443: The correct date for [5] is 2014.

Reference
---------
E. Hazan, A. Agarwal and S. Kale: Logarithmic regret algorithms for online convex optimization. Machine Learning Journal 69(2-3), 2007, pp. 169 - 192.
Summary: The paper provides an excellent contribution to the field of online learning and is more than likely to inspire a stream of future work.
Author Feedback
Author rebuttal: We thank the reviewers for their valuable and detailed comments, which will be certainly helpful for revising the paper.

Reviewer_34:
1. We want to emphasize that our paper does provide novel results. These include an adaptive and parameter-free algorithm that achieves \epsilon-regret with NO dependence on the number of actions N (and therefore can be applied to the case when N is infinity, i.e. general online convex optimization discussed in Section 4). This is the first such algorithm as far as we know. The closest work is NormalHedge by Chaudhuri, Freund and Hsu, 2009, but their regret has explicit dependence on N (see further comparison in the last paragraph of Section 3.3). We do not think this could be obtained by the usual "potential minimization techniques" that you referred to.

Other new results in this paper include a new boosting algorithm that is computationally faster since it ignores a large number of examples on each round, and at the same time still enjoys strong theoretical guarantees, i.e. exponentially dropping error rate (as AdaBoost) and optimal margin distribution bound (that AdaBoost fails to achieve).

2. "The reduction of..., but it is not really original": As far as we know, our reduction of several online learning problems into drifting games is original and has not been studied before. Potential functions have certainly been well-studied under different contexts, but using the potential of drifting games based on a relaxed minimax analysis to analyze online learning problems is a novel and useful approach, because it leads to results and algorithms that do not seem to be obtainable using the previously mentioned methods, as we already point out in Point 1. Other examples that demonstrate the power of this approach include:

a) A much simpler analysis we have for AbnormalHedge compared to the original complicated proof for NormalHedge.
b) A high probability analysis without resorting to any concentration results (Section 3.4).
c) The first relaxed minimax analysis for the multi-armed bandit problem (Section 4). (Note that the exact minimax analysis for this setting is still an open problem.)